# Psychedelic Targeting of Metabotropic Glutamate Receptor 2 and Its Implications for the Treatment of Alcoholism

**DOI:** 10.3390/cells12060963

**Published:** 2023-03-22

**Authors:** Kevin Domanegg, Wolfgang H. Sommer, Marcus W. Meinhardt

**Affiliations:** 1Institute of Psychopharmacology, Central Institute of Mental Health, Medical Faculty Mannheim, University of Heidelberg, 69115 Heidelberg, Germany; 2Bethanien Hospital for Psychiatry, Psychosomatics, and Psychotherapy Greifswald, 17489 Greifswald, Germany; 3Department of Molecular Neuroimaging, Central Institute of Mental Health, Medical Faculty Mannheim, University of Heidelberg, 69115 Heidelberg, Germany

**Keywords:** alcohol addiction, metabotropic glutamate receptors, serotonin 2a receptors, psychedelics, psilocybin, functional selectivity, epigenetics

## Abstract

Alcohol abuse is a leading risk factor for the public health burden worldwide. Approved pharmacotherapies have demonstrated limited effectiveness over the last few decades in treating alcohol use disorders (AUD). New therapeutic approaches are therefore urgently needed. Historical and recent clinical trials using psychedelics in conjunction with psychotherapy demonstrated encouraging results in reducing heavy drinking in AUD patients, with psilocybin being the most promising candidate. While psychedelics are known to induce changes in gene expression and neuroplasticity, we still lack crucial information about how this specifically counteracts the alterations that occur in neuronal circuits throughout the course of addiction. This review synthesizes well-established knowledge from addiction research about pathophysiological mechanisms related to the metabotropic glutamate receptor 2 (mGlu2), with findings and theories on how mGlu2 connects to the major signaling pathways induced by psychedelics via serotonin 2A receptors (2AR). We provide literature evidence that mGlu2 and 2AR are able to regulate each other’s downstream signaling pathways, either through monovalent crosstalk or through the formation of a 2AR-mGlu2 heteromer, and highlight epigenetic mechanisms by which 2ARs can modulate mGlu2 expression. Lastly, we discuss how these pathways might be targeted therapeutically to restore mGlu2 function in AUD patients, thereby reducing the propensity to relapse.

## 1. The Use of Serotonergic Psychedelics in the Treatment of Alcoholism

Alcohol is widely regarded as the most detrimental substance when considering both the harm to users and the harm to others [1]. Its abuse is responsible for 3 million deaths per year, accounts for 5% of all disability-adjusted life years and has an economic harm equivalent to 2.6% of the global GDP [2,3]. Alcohol is the leading cause for a broad range of disease- and injury-specific health burdens, including cancer, liver cirrhosis, tuberculosis, road injuries and self-harm [3]. The predominant diagnostic classifications used in the context of alcohol misuse are alcohol use disorders (AUD) and alcohol dependence, defined by DSM-5 and ICD-11, respectively, and is also simply termed alcoholism. It is estimated that 107 million people worldwide are diagnosed with AUD, with both the prevalence and treatment seeking rates showing significant country variations [4,5]. In Europe, approximately 1 in 12 people are estimated to be classified with AUD, with an expected treatment rate of 22% [5]. Many people with AUD engage in very heavy drinking (>100 or 60 g/day for males or females, respectively), which for Europe comprises around 1% of the population aged 15–65 years [6]. These people account for about half of all liver cirrhosis cases and have dramatically shortened life expectancy (>25 years) [6]. Thus, bringing down excessive consumption for harm reduction is an important public health goal. Treatment options for AUD include mutual-aid groups such as Alcoholics Anonymous, evidence-based behavioral treatments such as cognitive behavior therapy and motivational interviewing, as well as pharmacological treatments such as disulfiram, acamprosate, naltrexone and nalmefene [7]. Unfortunately, the approved pharmacological treatments for alcoholism are limited in their effectiveness, with only a minority of patients experiencing significant benefits [8,9,10]. Small effect sizes have also been observed for psychotherapy, raising the need for novel treatment approaches with robust effect sizes [10].

Recent findings suggest that the use of serotonergic psychedelics in combination with psychotherapy may be a promising treatment approach for a variety of psychiatric disorders, including major depression and substance use disorders such as AUD [11,12]. Serotonergic psychedelics are psychoactive drugs, including psilocybin, lysergic acid diethylamide (LSD), N,N-dimethyltryptamine (DMT) and 2,5-Dimethoxy-4-iodoamphetamine (DOI), that can elicit altered states of perception, cognition and emotion. A meta-analysis from six randomized clinical trials conducted in the 1960s and 1970s demonstrated that AUD participants treated with LSD are significantly improved at short-term (2–3 months) and medium-term (6 months) follow-up [13]. Moreover, a proof-of-concept study followed by a recent randomized controlled trial (RCT) demonstrated the effective usage of psilocybin for the treatment of AUD [14,15]. Two single doses of psilocybin in addition to the standard cognitive behavioral-based psychotherapy demonstrated a significant decrease in the percentage of heavy drinking days during a 32-week period [15]. These promising data, demonstrating a reduction in heavy drinking, encourage continued clinical research as well as a better understanding of the underlying molecular mechanism. 

The pharmacological, neural and psychological mechanisms by which psychedelics mediate their therapeutic effect remain largely speculative. A comprehensive review, recently published, summarizes and critically evaluates the current knowledge at these three different levels and outlines several potential directions to identify more specific mechanisms, and thus to allow a better targeted treatment [16]. To date, it has been discovered that psychedelics are partial agonists of brain serotonin receptors, inducing their behavioral responses mostly through the 5-hydroxytryptamine (serotonin) 2A receptor (2AR) [17,18,19]. Although crucial for their effect, substantial evidence over the last decade indicates that the 2AR alone might not fully explain the psychedelic-specific behavioral response and thus further physiological interactions must be considered [20,21,22]. One of the most promising targets is the metabotropic glutamate receptor 2 (mGlu2) that has been demonstrated to be necessary to induce the pharmacological and behavioral effects of psychedelics [23]. Interestingly, the 2AR can interact through specific transmembrane domains with the mGlu2 to form a functional complex that triggers unique intracellular responses in the presence of psychedelics [24]. mGlu2 has been extensively implicated in the pathology of AUD, and preclinical evidence demonstrated the potential of the mGlu2 modulation to treat alcohol relapse [25,26]. In this review, we use data from animal and human studies to assess the role of mGlu2 for behavioral and therapeutic responses elicited by psychedelics and consider these findings in the light of treating AUD. Specifically, we first introduce the role of mGlu2 in alcoholism before we then specify the mode of action of psychedelics, followed by a section that describes the interaction between 2AR and mGlu2 signaling. A special focus is then set on the cross-signaling of 2AR and mGlu2 through the formation of a receptor–receptor heteromer. We close this review with future directions and therapeutic implications of serotonergic psychedelics for the treatment of AUD.

## 2. The Role of the Metabotropic Glutamate Receptor 2 in the Pathology of AUD

AUD, or substance use disorder (SUD) in general (DSM-5), is considered a brain disease defined by the repeated and continuous use of legal or illegal substances that cause clinically significant impairment [27]. For AUD, this means a systematically biased choice preference for alcohol at the expense of healthy rewards and continued use despite adverse consequences [28]. Neuropathologically, a model of altered neuroplasticity has been proposed as an underlying pathological mechanism [29]. Since glutamate is an important mediator of synaptic plasticity, it is no surprise that glutamate receptors are highly involved in the development and progression of substance dependence and alcoholism in particular (see Box 1 for details of the glutamatergic system in the brain) [30,31,32].

Specifically, mGlu2 is highly abundant in the mesocorticolimbic and associated circuitries related to reward and drug seeking [33]. The prolonged exposure to drugs such as ethanol is believed to reduce mGlu2′s regulatory function in these systems, most notably in the medial prefrontal cortex (mPFC) and the nucleus accumbens (nAC), and by that contribute to the development of addiction-related behaviors [33]. For alcohol dependence in particular, a reduction of the GRM2 expression (the gene coding for mGlu2) was found in the anterior cingulate cortex (ACC) in patients with AUD as well as in the infralimbic cortex of rats with a history of alcohol dependence [34]. The role of mGlu2 in alcoholism was further strengthened by two studies in Grm2 mutant rats. First, it was demonstrated that the loss of functional mGlu2 in alcohol-preferring P rats due to the Grm2 cys407* mutation resulted in elevated alcohol consumption [35]. The second study discovered that the Grm2 cys407* mutation is also linked to excessive alcohol consumption in Hannover-derived Wistar rats [36]. Of note, one study did not find an association of prelimbic mGlu2 expression and high alcohol drinking [37]. Recently, we demonstrated a causal link between mGlu2 and AUD. Impaired mGlu2 function in the mPFC of rats resulted in excessive alcohol seeking and cognitive inflexibility, and these deficits were restored in postdependent rats—a well-established model of AUD [38]—by regional re-expression of the receptor [39].

Box 1The glutamatergic system in the brain.**Glutamate** is considered the primary excitatory neurotransmitter of the central nervous system and has an important role in all aspects of brain function including the modulation of synaptic plasticity [40,41]. The release and concentration of glutamate are tightly regulated to ensure proper function at its targets, the glutamate receptors. These are divided into ionotropic glutamate receptors (iGlu) and metabotropic glutamate receptors (mGlu). While iGlu such as N-methyl-D-aspartate receptor (NMDAR), kainate and α-amino-3-hydroxy-5-methyl-4-isoxazolepropionic Acid receptors (AMPAR) are ligand-gated ion channels that generate excitatory currents, mGlu are G-protein-coupled receptors (GPCRs) that activate second messenger signaling pathways and modulate synaptic transmission and plasticity on a longer scale [42]. The coordinated functions of iGlu and mGlu regulate the neural glutamate signaling, and dysfunctions or imbalances of these receptors, especially mGlu, have been implicated in various neuropsychiatric diseases [43,44].**mGlu** exist as constitutive dimers and can be further subdivided into class I, II and III [45]. Class II consists of mGlu2 and mGlu3, which are distributed in various areas of the brain and are mostly detected at the synapse of neurons [43]. There, they are primarily localized in presynaptic locations (excluding the active zone [46]), where they inhibit the release of the neurotransmitter glutamate and gamma-aminobutyric acid (GABA) in excitatory glutamatergic neurons and inhibitory GABAergic interneurons, respectively [45]. Apart from its presynaptic localization, mGlu2/3 is also found throughout the axon, at the postsynaptic membrane and on glia cells [46,47,48,49,50]. The precise localization of mGlu at the synapse determines its synaptic functioning and is dependent on differential mechanisms of trafficking and positioning [51]. Furthermore, species differences have been observed for the localization of mGlu2 and mGlu3: in the primate PFC, mGlu2 is the predominant presynaptic receptor and mGlu3 the predominant postsynaptic receptor, while in the rat PFC both receptors are predominantly found on presynaptic terminals [50,52].GPCRs mediate signaling through one of the four subfamilies of heteromeric G proteins (G_s_, G_i/o_, G_q/11_ and G_12/13_), which consist of an α, β and γ subunit. mGlu2/3 canonically binds to G_i/o_ and activates related signaling pathways, depending on the molecular localization.**The activation of presynaptic and postsynaptic mGlu2/3** generally results in opposite effects (Figure 1). Downstream signaling pathways of presynaptic mGlu2/3 include the inhibition of adenylyl cyclase and the protein kinase A via G_i/o_ α subunit, as well as the inhibition of voltage-gated calcium channels (VGCCs) and the activation of G protein-coupled inward rectifying Potassium channels (GIRKs) via G_i/o_ βγ subunits [53,54,55]. Together, these signaling pathways inhibit the glutamate release in presynaptic glutamatergic neurons and thus provide a feedback mechanism to prevent excessive excitation and induce long-term depression (LTD). Activation of postsynaptic mGlu2/3 on the other site results in G_i/o_-mediated NMDAR activation through various protein kinases including PKA, PKB, PKC and glycogen synthase kinase-3 beta (GSK-3B), as well as NMDAR trafficking by Snare proteins [56,57,58,59,60]. The activation of postsynaptic mGlu2/3 can further result in the increased surface expression of AMPAR via GSK-3B and extracellular signal-regulated kinases (ERKs) [61]. Taken together, postsynaptic mGlu2/3s potentiate the NMDAR current and regulate the NMDA/AMPA receptor trafficking, and thus cooperatively induce and maintain long-term potentiation (LTP). These findings demonstrate the sophisticated and opposing molecular mechanism of mGlu2/3 at presynaptic (LTD) and postsynaptic (LTP) locations required to ensure the correct and sustainable glutamatergic neurotransmission in the brain.

**Figure 1 cells-12-00963-f001:**
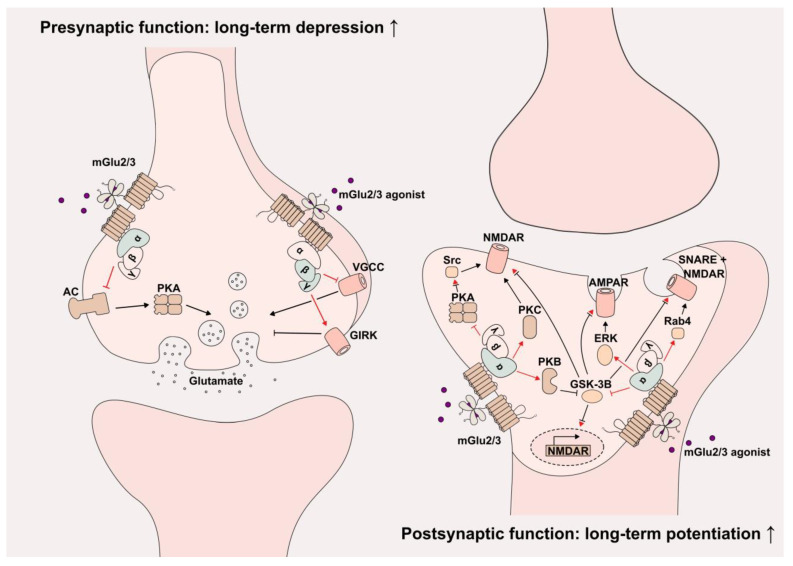
Molecular mechanisms of presynaptic and postsynaptic mGlu2/3 activation. Presynaptic (left) and postsynaptic (right) mGlu2 activation induces long-term depression and long-term potentiation, respectively. The relevant signaling cascades are displayed. Red indicates direct G-protein signaling consequences; red inhibitory arrow indicates second inhibition in the respective path. AC: Adenylyl cyclase, AMPAR: α-amino-3-hydroxy-5-methyl-4-isoxazolepropionic acid receptor, ERK: Extracellular signal-regulated kinases, GIRK: G protein-coupled inward rectifying potassium channels, GSK-3B: Glycogen synthase kinase-3 beta, NMDAR: N-methyl-D-aspartate Receptor, PKA: Protein kinase A, PKB: Protein kinase B, PKC: Protein kinase C, Rab4: Ras-related protein Rab-4, Src: Proto-oncogene tyrosine–protein kinase Src and VGCC: Voltage-gated calcium channels.

The virus-mediated rescue of mGlu2 in post-dependent rats was able to attenuate excessive alcohol seeking, suggesting the normalization of mGlu2 as a therapeutic opportunity [34]. AZD8529, a highly selective positive allosteric modulator (PAM) of mGlu2, was demonstrated to suppress cue-induced alcohol seeking responses in rats [62]. This effect was completely eliminated in rats lacking functional mGlu2, suggesting that the therapeutic effect is mediated by the activation of mGlu2 [62]. Furthermore, the administration of either the mGlu2 PAM LY487379 or one of the two mGlu2/3 agonists LY379268 and LY354740 to a well-established rat model of relapse demonstrated a significant decrease in relapse-like alcohol consumption [26]. This is in accordance with a prior study demonstrating reduced alcohol seeking and relapse behavior in alcohol-preferring rats after the administration of mGlu2 agonist LY404039 [63]. In our recent study, we demonstrated that craving and cognitive impairment can be rescued by restoring prefrontal mGlu2 levels [39]. In the same study, the administration of the psychedelic psilocybin was capable of restoring mGlu2 levels in alcohol-dependent rats. Given that psilocybin also reduced relapse behavior (REF), we hypothesize that the effect of the psychedelic on alcohol craving and drinking is, at least in part, mediated via mGlu2.

In sum, these studies demonstrate the pathological downregulation of mGlu2′s regulatory function in brain circuitries related to addictive behaviors and further pinpoint mGlu2 as a strong candidate for the treatment of AUD and preclinical trials targeting mGlu2 with agonists, and PAMs demonstrate promising results. Especially interesting in the light of the current clinical trials with psychedelics is the usage of psilocybin to target mGlu2 dysfunctions in alcohol-dependent rats. This molecular link between psilocybin and mGlu2 might at least partly explain the positive results observed in preclinical animal models and clinical studies for patients with AUD. The molecular mechanisms by which psilocybin and other psychedelics interact with or affect mGlu2 is only sparsely characterized, and further research is urgently needed for treatment development. In the next chapters, we summarize our current knowledge about the molecular pharmacology of psychedelics and describe their known interactions with mGlu2.

## 3. Molecular Pharmacology of Serotonergic Psychedelics

For a long time, psychedelics have rather been defined by their behavioral effect, more precisely by their ability to trigger altered states of consciousness, than by their mechanism of action. On a molecular level, most psychedelics are not very selective in their binding to receptors and demonstrate a strong variety in their potency and efficacy profiles among one another [64]. Nevertheless, overwhelming evidence in the last two decades pinpointed the 2AR as the central receptor for the action of psychedelics in humans, strengthening the pharmacological definition for serotonergic psychedelics as 2AR agonists [65,66,67]. It is widely believed that postsynaptic 2ARs located on dendrites of layer V pyramidal neurons are primarily responsible for the behavioral and therapeutic effect of psychedelics [66]. Interestingly, while some 2AR agonists induce the unique behavioral effects, others, though structurally similar, lack comparable psychoactive properties. Current research suggests that this phenomenon might be either explained by the concept of biased agonism or as recently shown, by the idea of location bias [68,69]. Box 2 summarizes 2AR-specific signaling cascades in the brain, that are important to understand the role of 2AR in the CNS, where it primarily increases the neuronal excitability and firing rate of glutamatergic and GABAergic neurons [70].

Besides differences in signaling pathways between psychedelic and non-psychedelic 2AR agonists, the activation of 2AR also shows distinct transcriptome responses dependent on the agonist [68]. In general, psychedelics rapidly induce the transcription of neuroplasticity-related genes [71]. One gene that shows a 2AR-dependent transcriptional change upon psychedelic stimulation is the brain-derived neurotrophic factor (BDNF), a growth factor that is well known to induce neural plasticity. The activation of 2AR via DOI and psilocybin results in increased BDNF expression [72], and the chronic BDNF exposure results in decreased 2AR protein levels [72,73].

Interestingly, the 2AR-mediated BDNF expression can be regulated by mGlu2 activity. The administration of the mGlu2/3 agonist LY354740 and mGlu2/3 antagonist LY341495 can dose-dependently repress and enhance the DOI-induced BDNF expression, respectively [74]. If targeted by psychedelics, including LSD, DOI and psilocybin, another unique cellular response is triggered: the induction of the early growth response 1 (Egr-1) and Egr-2 via G_i/o_ signaling and Src [24,34,68]. The inhibition of the tyrosine kinase Src completely abolished 2AR-mediated gene responses [68]. Egr genes encode transcription factors belonging to the class of immediate early genes that are also known to induce neural plasticity and by that contribute to the pathology of several addictions [75,76]. The regional-specific downregulation of Egr-1 and Egr-2, for instance, has been found in animal models of alcohol dependence [34]. Interestingly, the DOI-induced expression of Egr-2 was abolished in Grm2 knockout mice [23]. Furthermore, the Egr-1 was demonstrated to positively regulate Grm2 promoter activity [77]. The same authors additionally demonstrated that the binding of Egr-1 to the Grm2 promoter is decreased in 2AR knockout mice and that these animals display repressive epigenetic changes affecting histone H3 and H4 at the Grm2 promoter [77].

Box 22AR-mediated signaling cascades in the CNS.**2ARs belong** to the family of GPCRs and primarily bind to the G_q/11_ protein subfamily. Here, it is important to note that 2ARs are distributed throughout the body and their activation produces a variety of physiological responses. In this summary, we focus on 2AR-mediated signaling cascades in the central nervous system (CNS), particularly in cortical neurons.The **biased agonism**, refers to the phenomenon that a ligand can selectively activate some, but not all downstream signaling pathways of a receptor [78]. This has been described for 2ARs [68] and may explain differences in behavioral responses, which can be elicited upon the administration of psychedelic and non-psychedelic 2AR agonists.**Non-psychedelic 2AR agonists**, such as serotonin (5-HT) or tryptamine, preferentially activate canonical G_q/11_ signaling and its downstream effector phospholipase C (PLC) and mitogen-activated protein kinase kinase (MEK), resulting in calcium mobilization and the subsequent glutamate release, as well as the activation of various kinases, including ERK [79]. Intriguingly, the lack of the α subunit of G_q/11_ results only in a partial attenuation of a DOI-induced head-twitch response (HTR), a behavioral proxy in rodents for the psychedelic response induced by 2AR activity [80,81].**Psychedelic 2AR agonists** are able to activate additional signaling pathways apart from G_q/11_ signaling. A chimeric G protein-based assay recently proved that many GPCRs, including 2AR, interact with a much larger diversity of G protein subfamilies [82], The most significant coupling for 2AR, apart from canonical G_q/11_ proteins, was the interaction with G_i/o_ proteins. A previous study demonstrated the activation of G_i/o_ and its downstream effector Src in cortical cultures upon LSD [68]. However, the coupling of G_i/o_ α subunits to the 2AR remains questionable due to structural features that are expected to prevent their receptor binding, as discussed in [83]. Canonical G_q/11_ signaling, although neither unique nor specific, was, however, still required for the full behavioral response in mice induced by LSD or DOI [68]. Interestingly, G_q/11_ signaling shows an augmented response for DOI compared to the non-psychedelic 2AR agonist lisuride that differs in the phosphorylation magnitude of G_q/11_-cascade signaling proteins [84].Accessory **C terminus-dependent signaling proteins** are induced by GPCRs. In particular, the stimulation of 2AR by psychedelics results in the activation of G protein-independent mechanisms. Among these, is the postsynaptic density protein of 95 kDa (PSD-95) that can bind to the C-terminus of 2AR and by that seems to be involved in the in vivo expression of psychedelic drug actions. The absence of **PSD-95** nearly completely abolished the DOI-induced HTR and strongly reduced the phosphorylation of ERK and glycogen synthase kinase-3 beta (GSK-3B) [85]. The unphosphorylated (active) form of **GSK-3B** is able to phosphorylate PSD-95 to subsequently induce AMPAR mobilization and LTD [86]. Interestingly, 2AR antagonists can inactivate GSK-3B, while DOI demonstrated either no change in GSK-3B phosphorylation or surprisingly, like 2AR antagonists, an increase in phosphorylation [85,87]. The administration of DOI demonstrated a specifically strong increase in GSK-3B phosphorylation in mice with a β-arrestin2-knockout, suggesting that GSK-3B is further regulated through β-arrestin-2-mediated signaling [88]. **β-arrestin-2** is another key regulator involved in G protein-independent 2AR signaling that acts as a scaffolding protein and is known for its capacity to desensitize GPCRs. The release of endogenous serotonin activates β-arrestin-2-dependent signaling, while the psychoactive N-methyltryptamines as well as DOI activate the signaling independent of β-arrestin [89,90]. Additionally, the β-arrestin-2 knockout mice demonstrated no change in DOI-induced HTR and HTR tolerance, suggesting that the behavioral effects of psychedelics are β-arrestin-2-independent [91]. In another study, however, LSD displayed β-arrestin-biased signaling at 2AR, and β-arrestin-2 was required for LSD-induced behaviors in mice [83,92], complicating the role of β-arrestin-2 signaling for serotonergic psychedelics. The **biased phosphorylation of 2AR at Ser280** was demonstrated for psychedelic agonists that displayed reduced receptor desensitization and internalization compared to non-psychedelic agonists [93]. This phosphorylation is G_i/o_-independent and protein kinase C (PKC)-dependent, which is a downstream activator of G_q/11_-signaling [93]. Thus, psychedelics may specifically stabilize a 2AR conformation that allows Ser280 phosphorylation, which in turn could affect the β-arrestin-2-mediated receptor internalization. Recently, cryogenic electron microscopy images demonstrated that LSD stabilizes the inactive-state structure of 2AR [83]. The authors further demonstrate that this inactive state 2AR is β-arrestin-dependent. Taken together, these studies suggest that the activation of alternative 2AR signaling pathways in addition to the canonical 2AR (G_q/11_-mediated) signaling mediate the psychedelic response (Figure 2).The **location bias** describes another mechanism of functional selectivity, whereas the location of GPCR activation affects the signaling. Recently, the location bias has been connected to the effects of psychedelics by demonstrating that psychedelics can activate **intracellular 2ARs**, which then mediate therapeutically relevant signaling [69]. DMT and psilocybin, unlike 5-HT, can easily pass through the cell membrane due to their lipophilicity and activate the 2ARs localized at endosomes and the Golgi apparatus. By enabling the 5-HT access to intracellular 2AR, they were able to mimic the neuroplasticity-enhancing effects of psychedelics and induce an HTR in mice. These findings provide a new perspective to some of the aforementioned studies related to the concept of the biased agonism, and open further questions related to the role of 2AR localization and trafficking.

**Figure 2 cells-12-00963-f002:**
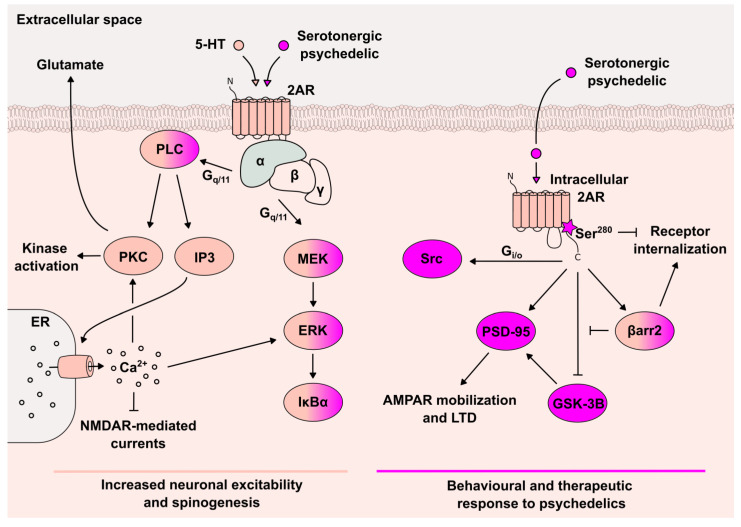
Canonical and psychedelic-related 2AR signaling pathways in neurons. Stimulation of 2AR by 5-HT (canonical agonist) results in the activation of G_q/11_ protein and the consequent activation of the PLC and MEK pathway (left). Together, these signaling pathways result in increased neuronal excitability and spinogenesis at the postsynaptic membrane. Stimulation of 2AR by serotonergic psychedelics regulate additional signaling pathways, including G_i/o_-mediated Src activation as well as G protein-independent pathways mediated by proteins such as PSD-95, GSK-3B and βarr2 (right). These signaling pathways, in addition to a biased phosphorylation of 2AR at Ser280, were demonstrated to be involved in mediating the behavioral response to psychedelics and are likely attributed to intracellular 2AR activation. Psychedelic-specific signaling is indicated in pink, while non-specific signaling is indicated in beige. AMPAR: α-amino-3-hydroxy-5-methyl-4-isoxazolepropionic acid receptor, βarr2: β-arrestin-2, ER: Endoplasmic Reticulum, ERK: Extracellular signal-regulated kinases, GSK-3B: Glycogen synthase kinase-3 beta, IκBα: Nuclear Factor of Kappa Light Polypeptide Gene Enhancer in B-cells Inhibitor, Alpha, IP3: Inositol Trisphosphate, NMDAR: N-methyl-D-aspartate receptor, PKB: Protein kinase B, PKC: Protein kinase C, PSD-95: Postsynaptic density protein 95, 5-HT: Serotonin and Src: Proto-oncogene tyrosine–protein Kinase Src.

## 4. Physiological Interaction between 2AR and mGlu2 Signaling and Its Implication for Psychedelics

Physiological interactions between 2AR and mGlu2 were demonstrated by a plethora of studies using electrophysiological, biochemical and behavioral evidence [65,94]. Early studies from Marek et al. demonstrated the overlap of the laminar distribution of 2AR and mGlu2/3 in the mPFC of rats using autoradiography [95]. Furthermore, they demonstrated that the stimulation of the 2AR increases the frequency and amplitude of spontaneous, glutamatergic excitatory postsynaptic potentials/currents (EPSPs/EPSCs) in L5p neurons of the mPFC, and that this becomes suppressed by the activation of mGlu2/3 [95,96]. This suppressant effect was in turn blocked by the mGlu2/3 antagonist, LY341495. Competition binding studies demonstrated increased 2AR and decreased mGlu2 agonist binding after the administration of mGlu2/3 agonists LY379268 and 2AR agonist DOI, respectively [24]. The binding of the mGlu2/3 agonist LY354740 was, however, not altered in the presence of DOI [97].

The psychedelic-increased L5p neuron activity has been further suggested to be linked to the head-twitch response (HTR) in rodents, a behavioral proxy for the hallucinogenic response, which non-psychedelic 2AR agonists such as lisuride fail to induce [81,98]. To investigate the interaction of mGlu2/3 with the 2AR on a behavioral level, Gerwitz and Marek measured the HTR in presence of a mGlu2/3 agonist and antagonist. In rats, the HTR induced by DOI was enhanced in the presence of the mGlu2/3 antagonist LY341495, while the mGlu2/3 agonist LY354740 suppressed the DOI-induced HTR [99]. The latter was further confirmed in mice and is in line with the electrophysiological 2AR-mGlu2 interaction in the PFC [100]. Furthermore, a mGlu2-selective positive allosteric modulator (PAM) was also capable of reducing the hallucinogenic-induced HTR [101]. Recently, a presynaptic crosstalk between 2AR and mGlu2/3 has been identified in the PFC of rats [102]. In an antagonist-like way, the two receptors were demonstrated to physiologically interact to ensure the correct glutamate exocytosis, with 2AR antagonists acting as indirect PAMs for the mGlu2/3. Similar results were demonstrated in the rat spinal cord glutamatergic terminals, where 2AR and mGlu2 were demonstrated to functionally interact in an antagonist-like fashion [103]. The necessity of mGlu2 in evoking the HTR was demonstrated in Grm2 knockout mice in two independent studies [23,104]. The number of head twitches in Grm2 knockout mice was strongly reduced compared to the wildtype: 25% and 37.5% of mice displayed the HTR after administration of 10 and 30 mg/kg DOI, respectively. Further behavioral evidence demonstrated that mGlu2 is also necessary for DOI to induce changes in the locomotor behavior [105]. These findings are intriguing and raise the question of how the presence of mGlu2 can be simultaneously necessary for the behavioral effects of psychedelics, while the antagonists of mGlu2 can enhance them.

Taken together, these studies suggest a predominantly antagonist-like interaction between the 2AR and the mGlu2 on a physiological level. They further indicate that mGlu2 is necessary to observe a full behavioral response to psychedelics and that mGlu2 agonists and antagonists can counteract and enhance the behavioral effects of psychedelics, respectively (Figure 3A). Here, it is important to mention that the increased EPSCs in L5p neurons and the observed HTR upon 2AR stimulation is not only attenuated by mGlu2 stimulation, but additionally by AMPAR blockage and the stimulation of other receptors such as NMDAR, 5-HT1A or 5-HT2C [65,106]. This indicates that the HTR, and perhaps the human behavioral and pharmacological response to psychedelics, is a 2AR-stimulated response that becomes modulated by a variety of signaling pathways, including but not limited to the physiological interaction of 2AR and mGlu2.

## 5. Cross-Signaling of 2AR and mGlu2 through the Formation of a GPCR Heteromer

On a molecular level, as discussed above, alternative 2AR signaling pathways seem to decide whether a substance can or cannot induce the behavioral response to psychedelics, with some studies suggesting the G_i/o_-related signaling as the most likely pathway. Interestingly, González-Maeso et al. attributed this G_i/o_ signaling to the mGlu2 receptor and its ability to form a heteromer with 2AR, and further demonstrated that the psychedelic-induced G_i/o_ activation, in particular, was significantly decreased in the presence of the mGlu2/3 agonist LY379268 [24]. Later, it was proposed that this 2AR-mGlu2 receptor–receptor complex can integrate the serotonergic and glutamatergic input, which then balances a G_i/o_- and G_q/11_-dependent signaling output [107] (Figure 3B). The authors further developed a metric, termed the balance index (BI), which is based on the change in G_i/o_ and G_q/11_ activity. Without the stimulation of either receptor, the 2AR-mGlu2 complex has a BI of 1.45 (reference BI). Depending on the input (agonist or antagonist), the output is shifted towards one of the two signaling pathways, thereby affecting the BI. mGlu2 antagonists as well as 2AR agonists such as serotonergic psychedelics promote high G_q/11_ to G_i/o_ activity, resulting in a BI smaller than the reference BI, while mGlu2 agonists and PAMs promote a high G_i/o_ to G_q/11_ activity, resulting in a higher BI than the reference BI [107]. The authors conclude that the balance between G_i/o_ and G_q/11_ signaling is crucial for understanding psychotic disorders and predicting the psychoactive behavior of pharmacological drugs.

Not only is it widely accepted nowadays that GPCRs act through a wide range of oligomeric states, it was demonstrated that the oligomerization can affect the pharmacology and function of its protomers [108,109]. Due to the often unique biochemical and functional signature of these complexes, disbalances in the heteromerization of GPCRs are suspected to play a role in several neurological disorders [110]. For instance, cocaine reward has been linked to the actions of the Adenosine 2A receptor–dopamine D2 receptor heteroreceptor [111]. Investigating the prevalence and function of GPCR heterodimers or higher-order structures in the native environment remains tricky to this day. Although numerous techniques have been developed, including biochemical, biophysical, physiological and computational methods, there is no definitive method available and every method has its own advantages and disadvantages [112,113]. In the next paragraphs, we will thus state and discuss the data regarding the existence and in vivo relevance of the heteromer proposed to mediate the behavioral and therapeutic response of serotonergic psychedelics and the 2AR-mGlu2 heteromer.

In 2008, it was demonstrated by co-immunoprecipitation that mGlu2 (but not mGlu3) forms a complex with the 2AR via its transmembrane domains 4/5 cortical pyramidal neurons by the use of mGlu2/3 chimeras with exchanged TM4 domains [24]. The heteromer formation of 2AR and mGlu2 was further confirmed using immunogold labeling, direct and sequential Förster resonance energy transfer (FRET) and SNAP/Clip-tag based homogenous time-resolved FRET methods, as well as the bioluminescence resonance energy transfer (BRET) [49,97,114,115]. It is important to note that some of these methods, such as FRET and BRET, were performed in heterologous systems that do not mirror the complexity of neurons in the brain (e.g., synaptic localization). Co-immunoprecipitation experiments performed from human, rat and mouse frontal cortices add further evidence for the constitutive physical association in native tissue [24,114,116]. Furthermore, the electron microscopy of the synaptic junction in the mouse frontal cortex displayed a close proximity between the two receptors, and the physical association between 2AR and mGlu2/3 has been demonstrated in the rat spinal cord [103,114].

Newer studies demonstrated that through the formation of the 2AR-mGlu2 heteromer, the stimulation of mGlu2 by LY379268 increased the G_q/11_-mediated Ca^2+^ release [49]. For this to happen, the coupling of G_i/o_ to the distal mGlu2 protomer (of the heteromer) is necessary [49]. The authors validated the biological relevance of this 2AR-mGlu2 heteromer-mediated signaling in the mouse frontal cortex and demonstrated that the mGlu2-dependent activation of G_q/11_ is altered in the frontal cortex from patients with schizophrenia [49]. The same group further identified three residues at the TM4 of the mGlu2 that are necessary for the heteromer formation [114]. Interestingly, it was demonstrated that the substitution of these residues prevented the heteromer formation and abolished psychedelic-like behavior induced by the psychedelic DOI [117]. Furthermore, they demonstrated that the 2AR is capable of affecting the localization and trafficking of mGlu2 through the formation of the heteromer [118]. The administration of DOI also increased the colocalization of the two receptors in Rab-5 positive endocytic vesicles, while the mGlu2 agonist LY379268 demonstrated the opposite effect [118]. Interestingly, high levels of intracellular 2AR (which are responsible for mediating the neuroplastic effects of psychedelics) colocalized with Rab5 in neurons [69]. These findings raise the question whether intracellular mGlu2 might be able to regulate psychedelic-induced signaling in Rab5-positive vesicles.

Experiments including the newly developed AlphaScreen assay and some of the techniques listed by Gomes et al. (e.g., affinity changes after ligand binding or bivalent ligands to demonstrate distinct heteromer-specific properties) are needed to further validate the 2AR-mGlu2 heteromer [112,117]. The same authors developed three criteria necessary to assess heteromerization in native tissue and concluded that the 2AR-mGlu2 heteromer only fulfills two out of three, which is still missing evidence that the heteromer can exhibit properties that are distinct from those of the protomers (criterion 2). They conclude that enough evidence is gathered to demonstrate the colocalization and interaction of the complex (criterion 1) as well as the loss of heteromer-specific signaling because of heteromer disruption (criterion 3). Another important question yet to be answered is whether the G proteins can simultaneously bind to the heteromer or whether they do it in a subsequential way.

Assuming that the complex exists and is necessary for the behavioral response of psychedelics, the question regarding the site of action remains. As mentioned before, mGlu2 is primarily found at presynaptic locations, but is also located postsynaptically and throughout the axon. 2ARs, on the other hand, are primarily recognized as postsynaptic receptors, although recent research suggests that they can be found at heterogenous synaptic locations including axon terminals of thalamocortical projections and hippocampal CA1 pyramidal neurons [119]. Studies from the mouse frontal cortex by Moreno et al. demonstrated that mGlu2 and 2AR were both found in the purified fractions of postsynaptic density (PSD) proteins and not in its presynaptic counterpart (PAZ), suggesting a postsynaptic localization of the complex [49]. The postsynaptic localization of the 2AR-mGlu2 heteromer would be in alignment with the observed activation of PSD-95 signaling after the administration of DOI [85]. Alternatively, other studies demonstrate that 2AR is also located presynaptically and that in rat spinal cords the 2AR-mGlu2 heteromer is located at the presynaptic sites [103,120]. The colocalization of 2AR and mGlu2 and their crosstalk was also confirmed in glutamatergic nerve endings from the prefrontal cortex of rats; however, no heteromeric complex was observed [102]. Taking into consideration that psychedelics are able to pass through the cell membrane and activate intracellular 2ARs, which are mostly located at the Golgi apparatus and in Rab5- and Rab7-positive endosomes, it is also plausible that the 2AR-mGlu2 heteromer can be found intracellularly [79].

Here, we reviewed data that indicate the existence of a heteromer that can integrate both the glutamatergic and serotonergic input and alter its downstream signaling accordingly. The 2AR-mGlu2 heteromer can shift the ratio of G_i/o_ and G_q/11_ signaling depending on the allosteric modulators bound to mGlu2 and 2AR, such as glutamate and serotonin, respectively. Patients with AUD often present changes in their glutamatergic and serotonergic systems, including the aforementioned alterations in the mGlu2 function [121,122]. These changes might result in suboptimal ratios of G_q/11_ to G_i/o_ signaling that can lead to pathological conditions such as those observed for patients with schizophrenia [24]. The administration of a 2AR inverse agonist (clozapine) in combination with a mGlu2 agonist (LY379268) was able to normalize G_q/11_ to G_i/o_ signaling in patients with schizophrenia and consequently reduce the antipsychotic-like behavior [107]. Thus, we propose that balancing G_i/o_ and G_q/11_ signaling (by targeting the 2AR-mGlu2 heteromer) might also be beneficial for a proportion of patients with AUD, and, more specifically, for those with alterations in mGlu2 activity.

## 6. The Role of Epigenetic Mechanisms in the Crosstalk between 2AR and mGlu2

AUD is a complex disorder with considerable influences from both genetics and the environment. Over the last decade, it has become clear that the environment can even affect the regulation of genes via an enduring mechanism, termed epigenetics. An epigenetic mechanism such as DNA methylation, histone modification and non-coding RNAs can regulate gene expression and thus also result in profound pathological effects. Several epigenetic modifications have already been linked to AUD and its associated behaviors, and many studies demonstrate promising results for the use of an epigenetic modifier as a therapeutic modality [123].

2AR and mGlu2 have been connected through an epigenetic mechanism in the mouse and human frontal cortex. Kurita et al. demonstrated that 2AR stimulation inhibits the expression of histone deacetylase 2 (HDAC2), which in turn results in an open/active mGlu2 promoter [124]. They additionally demonstrated that the treatment with the 2AR inverse agonists clozapine and risperidone reverses the effect and causes a high expression of HDAC2, which consequently results in a closed/inactive mGlu2 promoter [124]. The same group later discovered that the chronic treatment of clozapine decreases LY379268′s (mGlu2/3 agonist) ability to stimulate G-protein coupling and blunts LY379268′s anti-psychotic effects through the same HDAC2-dependent mechanism [125]. Co-administration of the HDAC inhibitor SAHA was able to conserve LY379268′s therapeutic abilities [125]. In cortical pyramidal neurons, specifically, it was demonstrated that the chronic treatment with clozapine and other atypical antipsychotics results in upregulated HDAC2 via the 2AR-dependent activation of NF-κB [126]. As shown in Figure 2, the ERK pathway downstream of the 2AR-mediated G_q/11_ signaling can activate IκBα, which is part of the IκB kinase (IKK) complex, a master regulator of NF-κB [127]. These studies demonstrate that treatment with 2AR inverse agonists affects mGlu2 expression through epigenetic mechanisms, more precisely through G_q/11_-mediated and HDAC2-dependent repressive histone modifications. Additional studies further attributed the antipsychotic effects of clozapine to the mGlu2 [105]. Future preclinical and clinical trials working with mGlu2/3 agonists or PAMs might thus consider the co-administration of HDAC2 inhibitors. In case future studies will demonstrate histone modifications at the mGlu2 in peripheral blood cells, another consideration might be to stratify patients based on their histone modifications at the mGlu2 promoter.

Ultimately, these findings also suggest that psychedelics as 2AR agonists might inhibit the expression of HDAC2 through 2AR-mediated signaling and by that upregulate the mGlu2 promoter activity. The activation of mGlu2 might then result in therapeutic effects, as described above. If experiments can proof this hypothesis, it might be worthwhile to use HDAC2 inhibitors in conjunction with psychedelics to enhance a possible therapeutic effect. Of note, the conditional knockout of HDAC2 pyramidal neurons in mice displayed a reduced HTR, suggesting the partial contribution for the behavioral effects of psychedelics, which leads to further questions [128].

Recently, it has become clear that psychedelics can cause chromatin remodeling: A single administration of the DOI induced persistent epigenetic changes in the neuronal nuclei of the frontal cortex of mice [129]. These changes in chromatin were especially significant in enhancer regions of genes involved in plasticity (e.g., enrichment of Egr1 and Egr2 motifs) and demonstrated a significant overlap with the genetic loci associated for psychiatric disorders [129]. The treatment with LSD was also demonstrated to upregulate genes involved in the epigenetic machinery such as Tet1, an enzyme involved in DNA demethylation [130]. Although lots of research is still needed to better understand the epigenetic landscape in regards to the treatment of alcoholism with psychedelics, we outlined some promising starting points, implicating mGlu2, which might result in relevant treatment strategies.

## 7. Future Directions and Therapeutic Implications

The dysregulation of glutamate signaling in the brain has been linked to several profound pathological conditions, including AUD. Due to mGlu2′s ability to regulate glutamate signaling through the modulation of presynaptic glutamate release and a variety of postsynaptic signaling pathways, mGlu2 has been considered a strong candidate for therapeutic interventions. Specific information about the role of mGlu2 and related treatment strategies for a wide range of neurodegenerative and neuropsychiatric diseases, including Alzheimer’s disease, schizophrenia and depression, can be found in a comprehensive review published by Li et al. [43].

Drug craving and relapse has been extensively linked to neurocircuitries associated with the nucleus accumbens, the medial prefrontal cortex and the anterior cingulate cortex, brain regions that show a high abundance of mGlu2 [131]. Together with the promising results of the aforementioned preclinical trials targeting mGlu2, it is thus no surprise that a continuous effort has been undertaken to use mGlu2 as a therapeutic target to treat alcoholism. The evidence so far suggests that prolonged alcohol intake reduces mGlu2′s regulatory function in these circuitries and that the targeting of mGlu2, either directly through mGlu2 agonists and PAMs, or indirectly through the psychedelic psilocybin, can restore the normal mGlu2 function and consequently rescue the pathological behaviors, most notably heavy drinking.

Preclinical and clinical trials for mGlu2 agonists and PAMs have now been performed for more than a decade; however, challenges in translating the promising results from animals to humans lead to not even one single FDA-approved treatment with mGlu2 modulators [43,132]. Apart from the large discrepancy between preclinical and clinical efficacy results, mGlu2 agonists have also demonstrated poor CNS penetration, excitotoxicity, receptor desensitization and tolerance after chronic administration, which questions their therapeutic validity [132,133,134]. However, newer studies using lower doses of mGlu2 agonists in a rat model of relapse demonstrated an effective reduction in alcohol consumption while displaying no signs of tolerance development [26]. The authors also demonstrate that mGlu2 PAMs are even more effective than agonists to restore control over relapse-like alcohol drinking [26]. mGlu2 PAMs generally show good safety and tolerability profiles, and although the first clinical studies failed to demonstrate a proof-of-concept, the interest in mGlu2 PAMs remains high [132]. Taken together, the direct activation of mGlu2 by agonists and PAMs might help to normalize mGlu2 function in patients with AUD, with mGlu2 PAMs being the more likely candidate to modulate mGlu2 directly and effectively in order to reduce alcohol seeking behaviors. However, proof-of-concept studies are urgently needed to advance the research of mGlu2 PAMs. Little is known so far about the effects of selective mGlu2 negative allosteric modulators (NAMs) in the context of addiction. Since several mGlu2 NAMs have been discovered in recent years, future studies will hopefully soon reveal whether they might be promising candidates alongside PAMs.

Apart from directly targeting mGlu2s with agonists or PAMs, our recent study demonstrated psilocybin’s ability to restore the mGlu2 expression and reduce the relapse behavior indirectly via 2AR signaling [39]. The single administration of either 1 or 2.5 mg/kg psilocybin was able to significantly reduce the relapse to alcohol compared to the vehicle (0 mg/kg), with no observed difference between the two doses [39]. Due to the heterogeneity of patients with AUD, we further proposed an FDG-PET biomarker approach to stratify the mGlu2 treatment-responsive individuals and pave the way towards personalized treatment. An experimental medicinal trial in patients with AUD as well as a cue-induced craving study are the logical next steps to demonstrate that a single administration of psilocybin can improve the cognitive flexibility and normalize functional brain connectivity, respectively. Other stratification biomarkers should also be considered, and ongoing trials should implement strategies such as the collection of biological samples for Omics-analysis as well as consider MRI and PET scanning.

Serotonergic psychedelics, in general, have been demonstrated to be potential candidates for the treatment of alcohol dependence. A systematic review including 20 human studies and 7 preclinical studies demonstrates overall promising data for psychedelics to effectively treat the symptoms of AUD, and the psilocybin was specifically demonstrated to be the most consistent compound [135]. The double-blind clinical trial for the treatment of AUD with psilocybin conducted by Bogenschutz et al. (NCT identifier number: NCT02061293) demonstrated the most convincing results so far, helping to consider proceeding with a phase 3 clinical trial. Further clinical trials using psilocybin (NCT05646303, NCT04718792, NCT05416229, NCT04141501, NCT04620759, NCT04410913, NCT01534494 and NCT05421065) are still to be completed and will give additional insights into its safety and efficacy. Apart from psilocybin, other psychedelics lack preclinical and especially clinical data to make any judgement about their effectiveness. Currently only one clinical trial for the treatment of AUD with psychedelics has been registered which does not use psilocybin, but in this case, uses LSD (NCT05474989). Taking into consideration that drugs of abuse share certain common effects, such as altering the brain’s reward pathways, it is reasonable to suggest that the use of psychedelics might yield beneficial effects for other SUDs as well. However, it is important to note that different drugs of abuse elicit distinct mechanisms in the brain that causes unique behavioral and physiological effects. Thus, only future double-blind, placebo-controlled RCTs will be able to answer whether this approach is useful for other SUDs. At the moment, clinical evidence suggests that the therapeutic use of psychedelics in the treatment of addiction is the most beneficial for AUD and tobacco-use disorder [136,137].

Most psychedelics, including psilocybin, are currently listed as Schedule I drugs under the United Nations 1971 Convention on Psychotropic Substances [138]. They are described as having a high potential of abuse, a lack of accepted safety and no accepted medical use. Preliminary data from clinical trials and additional research, however, demonstrate that psilocybin indicates low abuse and no physical dependence potential as well as a well-established physiological safety with rare psychological and psychiatric side effects [139,140,141]. If further trials with more rigorous methodologies can confirm these findings and clinical phase 3 studies can demonstrate efficacy, it seems likely that psilocybin and other psychedelics could be used for therapeutic purposes, including the treatment of patients with AUD. Compared to most other psychedelics, psilocybin has a pharmacokinetic profile that is more attractive for pharmacological treatment due to its shorter duration of action and faster elimination [139]. Recent pharmacokinetics and pharmacodynamics studies further strengthen the potential therapeutic use of psilocybin [142].

From a mechanistic perspective, it is important to understand how exactly psilocybin (or psychedelics generally) exerts its potentially therapeutic properties for alcohol dependence. Here, we outline data that points towards the metabotropic glutamate receptor 2, more precisely the normalization of mGlu2′s regulatory function through the stimulation of the serotonin 2A receptor. We demonstrate that psychedelics can induce additional 2AR signaling cascades apart from canonical signaling and result in distinct transcriptome responses. Furthermore, we demonstrate that mGlu2 and 2AR can regulate each other’s downstream signaling pathways and that this might be achieved through the formation of a 2AR-mGlu2 heteromer. We also highlight an epigenetic mechanism that links the 2AR to the mGlu2. Lastly, we demonstrate that psilocybin can restore cognitive flexibility in patients with AUD by upregulating the mGlu2 expression through 2AR. Several questions remain: Which of the 2AR signaling cascades induced by psychedelics are necessary for therapeutic effects and how exactly are they connected to mGlu2? Which of the 2AR-mediated transcriptional changes observed after the administration of psychedelics are affecting the mGlu2 function and are crucial for the therapeutic effect? Is the interaction between 2AR and mGlu2 only physiological or do they physically interact as a heteromer that displays heteromer-specific signaling in vivo, which is relevant for the pathology and treatment of AUD? Lastly, can psychedelics inhibit the expression of HDAC2 and by that modulate the mGlu2 function?

## 8. Conclusions

In summary, the current state of knowledge, despite the existing gaps, implies that psychedelics induce profound molecular changes via mGlu2, which are accompanied by circuit modifications that foster the improvement of AUD and challenge the efficacy of the currently available addiction pharmacotherapy. However, more work is needed to fully understand the exact molecular mechanism of psychedelics in AUD. Specifically, the application of state-of-the-art methods to tackle the above-mentioned open questions will provide useful insights for successful translational studies and treatment development.

## Figures and Tables

**Figure 3 cells-12-00963-f003:**
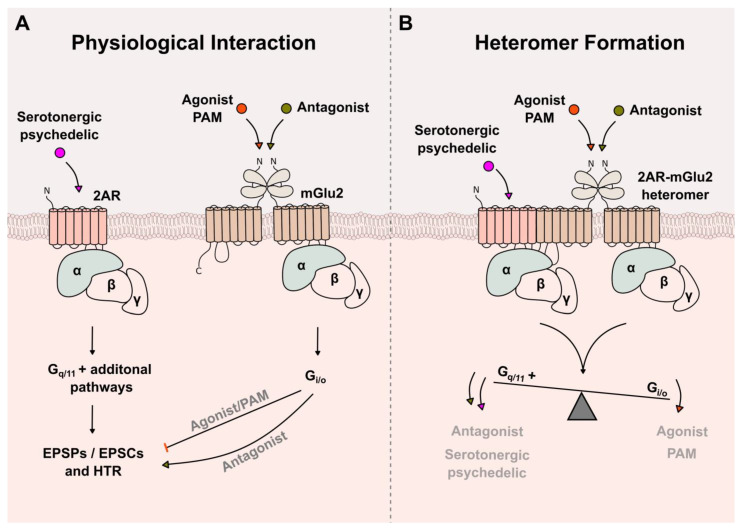
Cross-signaling of 2AR and mGlu2 through (**A**) physiological interaction and (**B**) the formation of a 2AR-mGlu2 heteromer. Activation of 2AR by serotonergic psychedelics induces EPSPs/EPSCs as well as psychedelic-related behaviors such as the HTR in rodents through the activation of G_q/11_ and additional signaling pathways (as described in Box 2). Stimulation of mGlu2 (by agonists or PAMs) or the presence of an mGlu2 antagonist was demonstrated to regulate these outcomes either (**A**) indirectly through its canonical G_i/o_ signaling or (**B**) directly through the formation of a heteromer with 2AR. The heteromer is assumed to integrate both serotonergic and glutamatergic input (such as serotonergic psychedelics and mGlu2 agonists, and PAMs or antagonists) and shift the balance of G_q/11_ + (and additional signaling pathways) to G_i/o_ signaling, accordingly. EPSC: Excitatory postsynaptic current, EPSP: Excitatory postsynaptic potential and PAM: Positive Allosteric Modulator.

## Data Availability

Not applicable.

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
