# Peer review of "Psychedelic Targeting of Metabotropic Glutamate Receptor 2 and Its Implications for the Treatment of Alcoholism"

_cells, 2023, doi:10.3390/cells12060963_

Round 1
Reviewer 1 Report
The present review by Domanegg et al. provides an up-to-date overview of psychedelic action that involves activation of their canonical target, serotonin 2a receptor, as well as the non-traditional and indirect target, mGlu2 glutamate receptor. The authors review behavioral, pharmacological, and cell signaling aspects of 5-HT2a-mGlu2 interaction as elicited by the administration of classic psychedelics and provide perspectives on how these psychedelics (acting in part via mGlu2) can be of utility to treat alcohol use disorder (AUD). Overall, this is an up-to-date review of an important, fast-developing topic that has recently received significant attention in both pre-clinical and clinical research. I only few minor comments:
- The authors should strongly consider adopting the current, IUPHAR-recommended nomenclature for the metabotropic glutamate receptors; that is mGlu2, not mGluR2. See: https://doi.org/10.2218/gtopdb/F40/2019.4 And adopt this nomenclature throughout the whole manuscript.
- In Box 1, and elsewhere as needed (e.g., p.12), the authors should cite an important work by Arnsten lab on the subsynaptic distribution of mGlu2 and mGlu3s in the monkey PFC (https://doi.org/10.3389/fnana.2022.849937; https://doi.org/10.1093/cercor/bhx005).
- I suggest replacing the sentence on Ln. 197-198. Especially the term “sophisticated” seems out of place.
- The authors should comment on whether the psilocybin-assisted therapy would be useful in SUD in general, or whether this approach would only benefit individuals with AUD.
Reviewer 2 Report
The article is well written and organized. The article is interesting and point out the pathways which might be a target therapy of AUD. The authors aim to indicate the targeting mGluR2 of serotonergic psychedelics in treatment of alcoholism. The authors address the physiological interaction between 2AR and mGluR2 in psychedelic-specific behavioural response. The authors present both in the molecular level and physiological interaction with several methodologies and the appropriated references. However, few details are warranted.
1.Line 196-199, the epigenetically effect or epigenetic change such as histone H3 and H4 could be addressed (according to reference 77).
2. According to the detail in the Box 2, the next activation of PSD-95 should be added in Figure 2.
3. Is there any reference BI level, if so, yield of the balance index should be indicated in line 284-288. This will show the clear idea of the cross-signaling of 2AR and mGluR2.
4. Figure Legend should be checked such as the symbolic letter (Gsk-b).
Reviewer 3 Report
Manuscript entitled “Psychedelic targeting of metabotropic glutamate receptor 2 and 2 its implications for the treatment of alcoholism” It is an interesting review, authors gathered all the information related to the mGluR2 and AR receptors in AUD. I will recommend this review for publication. However, I still have one suggestion.
If possible then authors should include a pictorial representation of these 2 receptors interaction and roles in disease conditions (AUD), although it is discussed in the review, it will be easy for understanding to the readers.
